# Changes of Viscoelastic Properties of Aptamer-Based Sensing Layers Following Interaction with *Listeria innocua*

**DOI:** 10.3390/s21165585

**Published:** 2021-08-19

**Authors:** Marek Tatarko, Sandro Spagnolo, Veronika Oravczová, Judit Süle, Milan Hun, Attila Hucker, Tibor Hianik

**Affiliations:** 1Department of Nuclear Physics and Biophysics, Faculty of Mathematics, Physics and Informatics, Comenius University in Bratislava, Mlynská dolina F1, 842 48 Bratislava, Slovakia; tatarko4@uniba.sk (M.T.); spagnolo2@uniba.sk (S.S.); oravczova2@uniba.sk (V.O.); 2Hungarian Dairy Research Institute Ltd., 9200 Mosonmagyaróvár, Hungary; jsule@mtki.hu (J.S.); mhun@mtki.hu (M.H.); ahucker@mtki.hu (A.H.)

**Keywords:** *Listeria innocua*, biosensor, quartz crystal microbalance, multi-harmonic analysis, DNA aptamer, viscoelastic properties

## Abstract

A multiharmonic quartz crystal microbalance (QCM) has been applied to study the viscoelastic properties of the aptamer-based sensing layers at the surface of a QCM transducer covered by neutravidin following interaction with bacteria *Listeria innocua.* Addition of bacteria in the concentration range 5 × 10^3^–10^6^ CFU/mL resulted in a decrease of resonant frequency and in an increase of dissipation. The frequency decrease has been lower than one would expect considering the dimension of the bacteria. This can be caused by lower penetration depth of the acoustics wave (approximately 120 nm) in comparison with the thickness of the bacterial layer (approximately 500 nm). Addition of *E. coli* at the surface of neutravidin as well as aptamer layers did not result in significant changes in frequency and dissipation. Using the Kelvin–Voight model the analysis of the viscoelastic properties of the sensing layers was performed and several parameters such as penetration depth, Γ, viscosity coefficient, *η*, and shear modulus, *μ*, were determined following various modifications of QCM transducer. The penetration depth decreased following adsorption of the neutravidin layer, which is evidence of the formation of a rigid protein structure. This value did not change significantly following adsorption of aptamers and *Listeria innocua*. Viscosity coefficient was higher for the neutravidin layer in comparison with the naked QCM transducer in a buffer. However, a further increase of viscosity coefficient took place following attachment of aptamers suggesting their softer structure. The interaction of *Listeria innocua* with the aptamer layer resulted in slight decrease of viscosity coefficient. The shearing modulus increased for the neutravidin layer and decreased following aptamer adsorption, while a slight increase of *µ* was observed after the addition of *Listeria innocua*.

## 1. Introduction

Foodborne diseases can be prevented in particular by rapid detection of pathogens in food [1,2]. Common sources of diseases in food are bacterial pathogens like *Listeria monocytogenes*, *Escherichia coli* O157:H7, *Staphylococcus aureus*, *Salmonella enterica*, *Bacillus cereus*, *Vibrio* spp., *Campylobacter jejuni*, *Clostridium perfringens*, and Shiga-toxin-producing *Escherichia coli* (STEC) [2,3,4].

Most of the diseases that originate from bacterial pathogens are caused by consumption of insufficiently heated or minimally processed food [5,6]. The consumption of insufficiently heated seafood is also of high risk because the minimal infective dose of pathogen from raw fish meat is usually relatively low (10–1000 CFU/mL) [5].

Currently, pathogen detection in industrial food laboratories is based on the identification of genetic material (DNA, RNA) of microorganisms using polymerase chain reaction (PCR). The immunological methods such as enzyme-linked immunosorbent assay (ELISA) and enzyme-linked fluorescent assay (ELFA) are also used as a standard (ISO norms). These methods have the advantages of shorter detection time, resulting in a significant increase in the number of inspected samples. Despite high sensitivity (up to 10 colony forming units (CFU)/mL), the remaining problem of these assays are relatively high cost and the requirement of qualified staff [7,8,9,10].

An alternative to the standard microbiological methods is biosensor technology. Biosensors consist of: 1. a bioreceptor that is sensitive to the analyte of interest, 2. a transducer that converts biological and chemical signal into electrical, optical, or gravimetric signals, and 3. the analyzer that provides quantitative information about the analyte [11,12]. The biosensors are usually classified by their transducer. The most common are optical, electrochemical, piezoelectric, and calorimetric biosensors [13]. A wide range of receptors is currently available such as antibodies, lectins, enzymes, nucleic acid, DNA/RNA aptamers, peptide-based aptamers or calixarenes [14].

Among receptors for biosensor development the DNA/RNA aptamers are of substantial interest. They are single-stranded DNA or RNA oligonucleotides produced in vitro by the SELEX method (systematic evolution of ligands by exponential enrichment) [15,16]. In solution, the aptamers fold into 3D structures forming a binding site for the target [16]. The affinity of aptamers to the target is comparable and can be even higher than those of the antibodies. The advantage of aptamers is the thermal stability, absence of immunogenicity, low-cost and rapid and reproducible production. They can be selected for a wide spectrum of targets such as small molecules, proteins, cells, bacteria, viruses, or tissues [16]. So far, the aptamers have been used in a variety of research contexts as a biorecognition element for detection of several species of bacteria [17,18,19,20,21], toxins [22,23], cancer cells [24], proteins [25] and other targets [26].

Among the biosensors, those based on acoustics principles are of substantial interest because they provide label-free detection of the analyte based on the analysis of the changes of resonant frequency. The basic element of acoustic biosensor is the piezoelectric transducer that generates acoustic waves following application of high-frequency voltage [27]. Several publications have already confirmed the effectiveness of the quartz crystal microbalance (QCM)-based detection of bacteria. Bayramoglu et al. [19] developed nanoparticle-mediated detection of *Brucella melitensis* in milk with a limit of detection (LOD) of 10^3^ CFU/mL. In the paper by Yu et al., a QCM aptasensor for *E. coli* with LOD of 1.46 × 10^3^ CFU/mL and with the detection time of 50 min was reported [20]. Such a biosensor included the application of rolling cycle amplification (RCA) method for production of multivalent, aptamer—based detection. Wang et al. [21] developed a QCM aptasensor for detection of *Salmonella typhimurium* with a LOD of 10^3^ CFU/mL and detection time of 60 min.

Detection of *Listeria* spp. by a QCM biosensor was reported only in few works, but none of them used the aptamer for detection of this pathogen. Vaughan et al. [28] detected *Listeria monocytogenes* using antibodies immobilized on the QCM surface with LOD of 10^7^ CFU/mL. However, this method suffered from blocking of some binding sites due to non-oriented immobilization of antibodies. Sharma and Mutharasan [29] reported sensors based on a piezoelectric cantilever, using antibodies for *Listeria monocytogenes*. They achieved LOD of 10^2^ CFU/mL in milk. However, such sensitivity was possible only after tertiary binding of antibodies to amplify the signal. Such a detection setup was rather complicated and time-consuming. In our recent work the biotinylated DNA aptamers immobilized at the neutravidin layers were used for detection of *Listeria innocua* using QCM. The obtained LOD was 1.6 × 10^3^ CFU/mL [17].

In the simplest interpretation of the results of QCM measurements the changes of resonant frequency are related to the changes of the mass of the sensing layers. However, according to Sauerbrey [30], this is valid only for rigid layers in a vacuum. In biosensors, the sensing layer is in a contact with liquid, therefore, the propagated acoustic wave dissipates when moving across the layer toward the liquid due to viscoelasticity. It is therefore important to analyze the viscoelastic properties of the sensing layers following interaction with bacteria.

There is increased interest in the application of multiharmonic QCM with dissipation (QCM-D) for analysis of viscoelastic properties of proteins or bacterial layers [31]. In most cases the focus was on the study of the adhesion of the bacterial colonies at the surface of the piezocrystal. It has been shown that simultaneous determination of the changes of resonant frequency, Δf, and dissipation, ΔD, is rather useful for the validation of the application of the Sauerbrey equation, for monitoring the adhesion and changes in cytoskeleton of the cells after external or internal stimuli and cytotoxicity (see [31,32] for review). QCM-D can be used also for determination of the protein layer thickness. It has been shown that QCM-D and atomic force microscopy (AFM) provide similar results [33]. QCM-D can be applied also for investigation of the bacteria fibrils and the cell adhesion [34].

In this work, we used multiharmonic QCM for analysis of the viscoelastic properties of the sensing layers formed by DNA aptamers at the neutravidin monolayer chemisorbed at the gold surface of the QCM transducer following interaction of bacteria *Listeria innocua* and *E. coli*. A portion of the results has been presented at the 1st International Electronic Conference on Biosensors [17]. However, the focus in this work was on the analysis of dissipation and viscoelastic properties of the sensing layers.

## 2. Materials and Methods

### 2.1. Chemicals

For the cultivation of bacteria and QCM measurements, the following chemicals were used: PBS (phosphate-buffered saline, 10 mM Na_2_HPO_4_, 1.8 mM KH_2_ PO_4_, 137 mM NaCl and 2.7 mM KCl diluted in MiliQ water with final pH 7.4), MiliQ water was prepared by a Millipore Milli-QElix^®^ Advantage 10 Water Purification System (Merck KGaA, Darmstadt, Germany). We also used TSA-YE (Triptone Soy Agar with yeast extract) and TSB-YE (Tripton Soy Broth with yeast extract) (Biokar Diagnostics, Allonne, France). Neutravidin (NA) was purchased from Thermo Fischer Scientific GmbH (Dreieich, Germany). The standard chemicals such as ethanol, NaCl, NH_3_ and H_2_O_2_ were purchased from Sigma-Aldrich (Steinheim, Germany). Biotinylated DNA aptamer of the sequence 5′-biotin-TAC TAT CGC GGA GAC AGC GCG GGA GGC ACC GGG GA-3′ was purchased from Eurogentec (Liege, Belgium). This aptamer (A15) has been developed for detection of bacterial pathogen *Listeria monocytogenes* [35]. However, according to the measurement of fluorescence intensity this aptamer modified at the 5′ end by fluorescein can bind, although with less affinity, the *Listeria innocua*. The latter is much less infectious in comparison with *Listeria monocytogenes*. Therefore, this bacteria has been used in our study.

### 2.2. Preparation of Bacteria Samples

The genus *Listeria* represents short Gram-positive rods 0.5–2 µm long with 1–4 flagella. They are aerobic or facultative anaerobic bacteria. According to somatic and flagellar antigens, *Listeria* spp. has up to 16 serovars due to its complex structure. Two types—*Listeria murrayi* and *L. grayi* have the same antigen ensemble, making them completely different from other types. Apart from *L. ivanovii*, there is no correlation between antigen structure and affiliation to a *Listeria* species. Some partial body antigens of *Listeria* are also common with other bacteria such as *staphylococci*, *enterococci* and *E. coli*, which may cause cross-reactions in serological tests. The genus *Listeria* is resistant to changes in the external environment and multiplies even at high salt concentrations (10% NaCl), which allows their long-term survival outside the host organism [36]. *Escherichia coli*, a member of Enterobacteriacae, is a Gram-negative, facultatively anaerobic bacteria typically rod-shaped 1.1–1.5 μm wide and 2–6 μm long. On the surface of *E. coli*, flagella, fimbriae or pili can be found. They have various strain-specific types of antigens presented on their cell wall derived from O lipopolysaccharides, flagella or capsular polysaccharides (K antigens). Usually, *E. coli* are serotyped based on the combination of O, H and K antigens, K antigens are often not stated, for example as in *E. coli* O157:H7, where O157 is somatic and H7 is a flagella antigens, respectively [37].

*Listeria innocua* B47 strain (strain No. GA_2018_10/01 (20A), isolated from raw poultry meat samples) was obtained from the culture collection of the Hungarian Dairy Research Institute Ltd. (HDRI Ltd., Mosonmagyaróvár, Hungary). *Escherichia coli* B40 strain (strain No. ATCC8739) was purchased from LCG standards Ltd. (Teddington, UK). The bacterial cultures were maintained at −80 °C in an ultra-low freezer (New Brunswick U410-86, Eppendorf AG, Hamburg, Germany) and retrieved from cryovials containing glycerol stock solution, and were subcultured twice in enrichment media. 16S ribosomal DNA of *L. innocua* B47 was genetically identified by Macrogen Europe (Meibergdreef 31, 1105AZ Amsterdam, The Netherlands). Consensus sequences were searched on the NCBI BLAST database (https://blast.ncbi.nlm.nih.gov/Blast, accessed on 11 August 2021). *Listeria innocua* is a non-pathogenic member of *Listeria* genus and presents similar habitats where *Listeria monocytogenes* occurs. The main difference between *L. innocua* and *L. monocytogenes* is that the first one is non-hemolytic [38]. On chromogenic media (ChromoCult^®^ Listeria agar acc. OTTAVIANI and AGOSTI, Merck KgaA, Darmstadt, Germany) *Listeria innocua* B47 produces typical green colonies without a halo [unpublished results of HDRI Ltd.] while *L. monocyogenes* green colonies has a halo [39]. The presence of a high number of non-hazardous *Listeria innocua* in food samples can influence the detectable number of foodborne-pathogen *Listeria monocytogenes*. Since *Listeria innocua* cells grow faster in selective enrichment broths than *L. monocytogenes* there is a high risk of giving false negative results for the latter pathogenic bacteria [39,40]. Pure bacterial cultures were inoculated with an inoculating lop on a Petri dish with TSA-YE solution. This was followed by incubation for 24 h at 30 °C. After growth of visible colonies, a sample was harvested and dispersed in TSB-YE broth (a), followed by incubation for 12–18 h. For further measurements, the bacterial solution was centrifuged and then diluted in PBS (b) to the desired concentration for the experiment.

The CFU/mL was calculated using formula (1) [41], for which information on the number of live bacteria capable of forming colonies at different dilutions on special agar is required. The procedure was as follows: solution (a) was diluted by decimal dilution from 10 up to 10^8^. Individual dilutions were applied (0.1 mL) each to 3 different TSA-YE agars (Agar A; B; C). This was followed by incubation for 24 h, after which the colonies were counted. Optical density value was determined as 0.1 OD at 600 nm.
(1)$$\frac{\mathrm{CFU}}{\mathrm{mL}}=\frac{\mathrm{Average}\;\mathrm{number}\;\mathrm{of}\;\mathrm{colonies}}{\left(\mathrm{Dilution}\right)*\left(\mathrm{Amount}\;\mathrm{of}\;\mathrm{inoculted}\;\mathrm{samples},\;\mathrm{mL}\right)}$$

### 2.3. Preparation of the Aptamer Layers

In experiments, the piezocrystals with fundamental frequency of 8 MHz (Total Frequency Control, Storrington, UK) were used. Both sides of the crystal had thin gold layers serving as electrodes. The working surface of the crystal has a circular shape of an area of 0.2 cm^2^. Prior aptamer immobilization the crystal was carefully cleaned with basic Piranha solution (29% NH_3_, 30% H_2_O and H_2_O_2_ with volumetric ratio 1:5:1, respectively) for 25 min. After this treatment, the crystal was washed three times with deionized water and stored in ethanol. After drying in a flow of nitrogen the crystal was placed in an acryl flow cell connected to the syringe pump (Genie Plus, Kent Scientific, Torrington, CT, USA). Then 125 μg/mL of neutravidin (NA) dissolved in deionized water was added at one side of the crystal with flow rate of 50 µL/min for approx. 30 min. Neutravidin is deglycolized avidin that contains SH groups that allow chemisorption at the gold surface. This process was controlled by measurements of the resonant frequency of the crystal. As soon as the frequency stabilized, the crystal was washed by deionized water to remove weakly adsorbed neutravidin molecules. The washing by PBS then followed. Finally, the biotinylated aptamers dissolved in PBS in a concentration of 0.5 µM have been added with the same flow rate during 30 min. Due to the high affinity of biotin to the neutravidin a self-assembly aptamer monolayer was formed [42,43,44].

### 2.4. Study of the Interaction of Bacteria with Aptamer Layers

Thanks to simple configuration and sensitivity, the QCM is among most widely used methods in the development of affinity biosensors. To ensure a minimum dependence of the crystal oscillation on the temperature, the AT-cut quartz crystals were used (cut at an angle of 35°15′ to the *Z*-axis) [45].

The principle of the QCM consists in measurements of the changes in the resonant frequency that are related to the changes of the mass at the crystal surface. According to Sauerbrey [30], the changes in the resonant frequency, Δ*f*, of the quartz crystal in vacuum are related to the changes of mass, Δ*m*, by equation:(2)$$\Delta f=\frac{-2nf_{0}^{2}\Delta m}{A\sqrt{\mu_{q}\rho_{q}}}$$
where *n* is harmonic number, *f*_0_ is the fundamental resonance frequency, *A* is effective crystal area, *μ_q_* = 2.947 × 10^11^ g·cm^−1^·s^−2^ is the shear modulus of elasticity and *ρ_q_* = 2.648 g·cm^−3^ is the crystal density.

In a water environment, the frequency can also be affected by viscous forces, therefore an additional term should be added to the Sauerbrey equation:(3)$$\Delta f=2f_{0}^{\frac{3}{2}}\sqrt{\frac{\eta_{L}\rho_{L}}{\pi \mu_{q}\rho_{q}}}$$
where *η_L_* is the viscosity and *ρ_L_* the density of the liquid, respectively [46].

The acoustic waves in a QCM transducer are generated by applying a high-frequency voltage to the electrodes sputtered at both sides of the crystal [47]. Depending on the thickness of the crystal, the fundamental resonant frequency is defined, mostly in the range of 5–30 MHz [47]. When the acoustic wave transits from the electrode to the adsorbed layer, it causes energy dissipation corresponding to observed phase shift and an attenuation. As the oscillation is modelled by the Butterworth–Van Dyke equivalent electric circuit, this attenuation can be estimated by motional resistance *R_m_*. Corresponding decay of the acoustic wave is characterized by penetration depth (decay length) Γ, that can be expressed as:(4)$$\Gamma =\;\sqrt{\frac{2\eta_{L}}{\omega \rho_{L}}}\;$$
where $\eta_{L}$ is liquid viscosity, ω is circular frequency of the oscillations and $\rho_{L}$ is the density of the liquid. Another factor related to penetration depth is dissipation factor, *D*, expressed as:(5)$$D=\frac{2\Gamma }{f_{0}}$$

The QCM experiments were performed using the computer-controlled Sark 110 vector analyzer (Seeed, Shenzhen, China). The device allowed measurement of fundamental and higher harmonic frequencies. The frequency changes increased linearly with the harmonic number, *n* (see Equation (2)). All measurements were performed in a flow mode. The bacterial suspension of the concentration determined by the standard method (see Section 2.2) was added to the crystal surface covered by aptamers with a flow rate of 50 µL/mL. The changes of fundamental and higher harmonic frequencies were continuously monitored. All experiments were performed at ambient temperature at around 20 °C with accuracy of ±0.5 °C. For apparatus setup see Dizon et al. [48].

By measurement of the frequency and dissipation it is possible to estimate the viscoelastic properties of the surface layer at the piezocrystal. However, it is useful to verify whether the analysis properly reflects the viscoelastic properties. For this purpose, the analysis of the normalized frequencies for more overtones *f_n_*/*n* and their relative changes Δ*f_n_*/*n* are helpful. As more of these values differ from each other, the higher is the viscoelastic component of the sample. For estimation of the viscoelastic parameters, it is also important to determine the dissipation changes. In general, small changes in dissipation up to 10^−6^ don’t imply the presence of prevalent viscoelastic phenomena in the sample, except for small viscoelasticity variations probably due to the interaction of the water with the layer. Saftics et al. [49] have proposed a critical number of the dissipation change, 2 × 10^−6^, as a limit for assuming a layer as viscoelastic or rigid. Greater numbers may suggest that the layer has a significant viscoelastic behavior.

For a study of the viscoelastic properties of the sample, it is useful to analyze the thickness of the hydrated layer, since water molecules are an integral part of this system. It is possible to calculate the thickness $h_{0}$ of the crystal using the resonant frequency, *f*_0_, at the beginning of the measurements according to the equation:(6)$$f_{0}=\frac{u}{2h_{0}}\to h_{0}=\frac{u}{2f_{0}}$$
where *u* is the velocity of the acoustic wave in the crystal (*u* = 3336 m/s). Once the thickness of the crystal has been calculated and the frequency shift obtained, the thickness $h_{1}$ of the adlayer can be calculated, using the equation:(7)$$\Delta f=f_{1}-f_{0}=\frac{u_{f}}{2\left(h_{0}+h_{1}\right)}-\frac{u}{2h_{0}}$$

Using this equation, it is also possible to calculate the thickness of subsequent layers at the piezocrystal surface. Considering that the contribution of the layer in the variation of the acoustic wave along the crystal is practically negligible, then $u_{f}\;$= u [50].

By means of the viscoelastic analysis it is possible to obtain information about the characteristics of the sensing layer, such as stiffness, viscosity, elasticity and loss moduli. Furthermore, the formation of crosslinking bonds or the possibility of freedom of movement of the molecules constituting the adlayer can be analyzed. The Kelvin–Voigt viscoelastic model has been developed for this purpose. Using this model, it is possible to calculate the viscoelastic properties of the adlayer considering the differences in frequency and dissipation at the different harmonics. This model was described mathematically by Voinova et al. [51], according to which the viscoelastic properties of the film are related to the variations in frequency and dissipation by the equations:(8)$$\Delta f\approx -\frac{1}{\left(2\pi \rho_{0}h_{0}\right)}\left[\left(\frac{\eta_{3}}{\Gamma_{3}}\right)+h_{1}\rho_{1}\omega -2h_{1}{\left(\frac{\eta_{3}}{\Gamma_{3}}\right)}^{2}\left(\frac{\eta_{1}\omega^{2}}{\mu_{1}^{2}}+\omega^{2}\eta_{1}^{2}\right)\right]$$
(9)$$\Delta D\approx \frac{1}{\left(\pi f_{n}\rho_{0}h_{0}\right)}\left[\left(\frac{\eta_{3}}{\Gamma_{3}}\right)+2h_{1}{\left(\frac{\eta_{3}}{\Gamma_{3}}\right)}^{2}\left(\frac{\eta_{1}\omega }{\mu_{1}^{2}}+\omega^{2}\eta_{1}^{2}\right)\right]$$
where $\Gamma_{3}$ is the decay length of the shear wave in the liquid; $\rho_{3}$ and $\eta_{3}$ are the density and viscosity of the liquid (water or very dilute saline solutions have approximately the same density, 0.9982 g/cm^3^, and dynamic viscosity, 1.0016 mPa·s, at 20 °C), $\rho_{0}$ and $h_{0}$ are the density and thickness of the quartz crystal, respectively 2.648 g·cm^−3^ and 0.208 mm for the crystal with fundamental frequency of *f*_0_ = 8 MHz (the thickness can be precisely calculated using Equation (6)), $h_{1}$, $\mu_{1}$, $\eta_{1}$ and $\rho_{1}$ are the thickness, the elastic shear modulus, viscosity and density of the adsorbed film; ω is the angular frequency of the oscillation.

The software used in this work based on Python has elaborated a viscoelastic model by means of these equations and provides data in accordance to the frequency and dissipation variation inserted in the model.

## 3. Results and Discussion

### 3.1. Formation of Aptamer Layer

In the first series of experiments, we measured adsorption kinetics of neutravidin on gold surface and those of biotinylated aptamers to the neutravidin layer. The changes in frequency, Δf, and dissipation, ΔD, were measured in long time kinetic in order to check the stability of the measuring values. Clean crystal mounted in an acryl flow cell was first washed with deionized water for several minutes in a flow mode until the resonance frequency stabilized. The addition of neutravidin dissolved in deionized water with a concentration of 125 μg/mL resulted in a sharp decrease of the fundamental and harmonic frequencies (Figure 1A). According to the recommendation from the producer, neutravidin was dissolved in deionized water, because it is not directly soluble in PBS. Washing the surface with deionized water resulted in a significantly smaller increase of the frequency, which corresponds to the removal of the weakly adsorbed neutravidin molecules from the surfaces. The resulting frequency change due to neutravidin adsorption determined from 5 experiments were Δ*f_s_* = −203.0 ± 18.3 Hz. We determined also dissipation changes, ΔD, that reflect the viscosity contribution. It can be seen from Figure 1B that addition of neutravidin also causes an increase of dissipation with an average value Δ*D* = (2.35 ± 0.49) × 10^−6^.

This confirms that the neutravidin layer is rather rigid and viscosity contribution is not significant. This result agrees well with our previous work [52] in which it has been shown that formation of the neutravidin layers is accompanied only by small changes in motional resistance, R_m_, which is a certain analogue of the dissipation. Thus, the neutravidin layer can be considered as a rather rigid structure, allowing the Sauerbrey equation to be applied for determination of the surface density of neutravidin molecules (see below).

After formation of the neutravidin layer, the surface was washed by PBS. This is because aptamers were dissolved in PBS and the changes in ionic composition can affect the surface properties of the layer and therefore also the resonant frequency and dissipation. This effect can be observed in Figure 1. These changes can be attributed to the influence of ionic strength on the surface viscosity [53]. Addition of aptamers dissolved in PBS in a concentration of 0.5 µM resulted in a decrease of the resonant frequency Δ*f* = −82.0 ± 2.1 Hz and in an increase of dissipation Δ*D* = (5.1 ± 1.1) × 10^−6^. The increase of dissipation is evidence of a significant contribution of the viscosity. After addition of the aptamer the washing of the surface by PBS caused only a slight increase in frequency, suggesting that a stable aptamer layer was formed on the surface of the neutravidin layer. Based on resonance frequency changes after application of neutravidin and aptamers, it is possible to determine surface mass density using the Sauerbrey equation as 5.6 × 10^12^ molecules of neutravidin and 1.08 × 10^13^ molecules of aptamers per cm^2^. Because neutravidin has four binding sites for biotin and two of them become unavailable due to their immobilization on the crystal surface, the total number of available neutravidin binding sites is 1.12 × 10^13^ per cm^2^. This means that the number of immobilized aptamer molecules is almost identical with available binding sites. However, this is only a rough estimation since adsorption of aptamers is accompanied also by increase of dissipation (Figure 1B).

### 3.2. Interaction of the *Listeria innocua* and *E. coli* with Neutravidin and Aptamer Layers

As we showed above, the aptamers cover practically all the surface of the neutravidin monolayer at the QCM transducer. However, in order to exclude possible non-specific interaction of bacteria with the neutravidin layer we studied the interaction of *Listeria innocua* and *E. coli* with only the neutravidin layer. Corresponding kinetics curves for frequency and dissipation changes following the addition of *Listeria innocua* are presented on Figure 2. It can be seen that *Listeria* did not cause significant changes in frequency and dissipation. A similar result has been obtained also for interaction of *E. coli* with neutravidin monolayers (Appendix A).

In the study of specific interaction of *Listeria innocua* with sensing layer we ensured the stability of the resonant frequency within 1–2 h under the control of the temperature 20 ± 0.5 °C. As soon as the frequency stabilized, the *Listeria innocua* was added step-wise at the sensing surface. The kinetics of the frequency changes following the addition of *Listeria innocua* at the surface of the QCM aptasensor is presented in Figure 3A. Addition of bacteria in the range of 5 × 10^3^–10^6^ CFU/mL resulted in the frequency decrease and in a slight increase of dissipation (Figure 3B). It is also seen that the noises in the frequency changes are around 2 Hz, which is comparable with mean frequency changes following the addition of the smallest applied concentration of *Listeria innocua*, 5 × 10^3^–10^4^ CFU/mL. Despite the tendency of frequency decrease and dissipation increase being obvious, the question arises as to whether this decrease is due to specific interaction of the bacteria with the sensing surface or due to a certain drift of the baseline. However, as can be seen from long-time kinetics of the frequency and dissipation changes (Figure 1), these parameters are rather stable. In the case of the frequency we observed only a slight increase of this value. Only after 240 min. of measurements did the slight decrease of frequency occur (Figure 1A).

Thus, the frequency and dissipation changes are due to specific interaction of the sensing layer with *Listeria innocua.* Comparing the changes in frequency at the lowest (5 × 10^3^ CFU/mL) and highest concentration (10^6^ CFU/mL) of *Listeria innocua* using Student’s *t*-test it can be concluded that these values are significantly different with *p* < 0.001. The changes in the frequency of the aptasensors following specific interaction with *Listeria innocua* have been analyzed quantitatively in our recent paper [17]. The frequency changes were relatively low, however similar values were observed also in experiments by Yu et al. [20] for *E. coli* detection by QCM. One of the possible reasons for the small changes of the frequency can be due to substantially lower penetration depth of the acoustic wave (approximately 120 nm, see Section 3.3) in comparison with the thickness of the bacterial layer (approximately 500 nm). Thus, the acoustics sensor “does not see” all the adsorbed bacterial layer but mostly only the part that closely contacted with aptamers [54]. In our previous work [17] we also determined the LOD = 1.6 × 10^3^ CFU/mL. The obtained LOD shows sufficient ability of the aptasensor to detect *Listeria innocua.* The sensitivity of detection is comparable with an infection dose of *Listeria monocytogenes*, that is similar to *Listeria innocua* and requires only 10^3^ CFU/mL concentration to cause foodborne illness [55]. The LOD obtained by the QCM method is better in comparison with a SERS-based immunosensor [56] for which the LOD of 10^4^ CFU/mL for *L. innocua* was determined.

In order to further check the specificity of the interaction of *Listeria innocua* with the sensing layer we performed also experiments with *E. coli* as a control, which should interact with aptamers to less degree [35]. *E. coli* has identical partial antigens with 13 subspecies of *Listeria* spp. (including *Listeria innocua* and *Listeria monocytogenes*) and causing cross-reaction in serological tests. Formation of the aptamer layer was identical to those presented above.

Figure 4 shows kinetics of the changes of the frequency and dissipation following the addition of *E. coli* in a concentration range of 10^2^–10^6^ CFU/mL. As can be seen stepwise addition of *E. coli* caused only a slight increase of the resonant frequency. However, addition of *E. coli* at concentration above 5 × 10^4^ CFU/mL caused an increase of dissipation for fundamental harmonics.

Thus, a rather small increase of frequency suggests that that *E. coli* did not interact specifically with the aptamer surface. However, an increase of the dissipation at the fundamental frequency was evidence of possible weak non-specific interaction of the *E. coli* with aptamer layer. Bacterial adsorption can be considered as a coupled oscillators. Depending on the stiffness of the spring connected the bacteria with the sensing surface, the frequency shift can be more or less positive (See Appendix A for explanation). In comparison, *Listeria innocua* caused frequency decrease already at 5 × 10^3^ CFU/mL. This result is surprising considering that the aptamer used had only slightly less affinity to *E. coli* in comparison with *Listeria innocua* [35] as determined by the fluorescence method. At the same time the interaction of bacteria with the surfaces is promoted also by release of the extracellular polymeric substance (EPS), which can cause non-specific interactions. However, as has been shown by Olsson et al. [57], EPS is more strongly connected with the QCM substrate and increases the mass loading. Therefore, EPS can contribute to the negative frequency shift. At higher concentrations of bacteria, the effect of elastic–spring coupling described above can promote the frequency increase.

One can also mention that occasionally during application of the higher concentrations of bacteria (above 5 × 10^5^ CFU/mL) we observed an increase of the resonant frequency and a decrease of dissipation (Appendix A). Because the samples contain live bacteria, we initially assumed that increased load caused removal of bacteria or even partial removal of the aptamers. However, based on the QCM-D measurements, Olsson et al. [57] suggested that bacteria cells on the protein surface behave as an elastic spring-like system, which caused an increase in frequency. This phenomenon was described as a coupled-resonator model, where adhering mass is coupled with resonator via adhesive “spring”, causing the restoring force on the crystal [58]. This does not mean that cells are completely detached from the surface, but there is a possibility that they are stretched towards the solution and their whole mass does not affect the resonant frequency. The scheme of this process is presented on Appendix A. Changes in dissipation are more important for these situations together with more complex evaluation of viscoelastic properties. However, further study of this phenomenon should be performed using alternative methods such as ellipsometry, AFM or electrochemical measurements. The data obtained from QCM measurements on the increase of the resonant frequency at relatively high concentrations of bacteria (5 × 10^5^ CFU/mL) were not suitable for the construction of the calibration curve. However, we used these data in later analysis of the viscoelastic properties presented below.

### 3.3. The Analysis of the Viscoelastic Properties of the Sensing Layers Using Kelvin–Voigt Model

Following the QCM analysis and the determination of the frequency and dissipation changes, it was possible to evaluate how the viscoelastic properties of the system change. The software developed in Python and based on the Kelvin–Voigt model has allowed different viscoelastic parameters to be obtained such as penetration depth, Γ_3_, viscosity coefficient, *η*, and the shear modulus, *μ*. For this purpose the data on the changes of frequency and dissipation for fundamental and up to 7th harmonics (3rd, 5th and 7th) were crucial. While the acoustics wave of the fundamental frequency penetrates deeper into the liquid, the higher harmonics penetrate only into to the densely packed layers. Therefore, fundamental frequency is useful for the construction of calibration curve, while higher harmonics are suitable for analysis of viscoelastic properties of the sensing layers [32].

The penetration depth describes the distance travelled by the evanescent acoustic wave which leaves the crystal and propagates in the medium. This length varies according to the properties of the medium and of the film adsorbed on the crystal (in its absence, the penetration depth for a crystal with 8 MHz fundamental frequency is about 200 nm in a medium such as water [59]). The penetration depth is also useful for understanding the maximum thickness of the sensing layer for which it is still possible to calculate the viscoelastic properties.

In Figure 5A, the trend of the penetration depth during the measurement is showed. Following adsorption of neutravidin (NA), the penetration depth decreased from about 250 nm to 120 nm. This is evidence that that neutravidin forms a very compact layer. Subsequently, the bond of the aptamer with neutravidin did not cause a significant variation in penetration depth. This may be because the nucleic acid aptamers have a looser structure and do not form an additional compact layer. Even incubation with *Listeria innocua* at different concentrations did not cause a significant decrease in the penetration depth (Figure 5B). This can be explained by the fact that the bacteria did not form a compact layer and did not cover completely the entire sensor area, but rather they bind to the aptamer in low numbers (this can be an additional reason for low frequency changes following the adsorption of bacteria). Finally, the last two bacterial incubations produced a slight increase in the penetration depth, reaching values comparable to those before interacting with aptamers. This could be explained by possible stretching of the bacterial cells toward the solution. Therefore, their whole mass does not affect the resonant frequency as discussed above in Section 3.2.

In any case, considering that the evanescent wave propagates for about 120 nm in the medium it is still possible to evaluate the variations in the viscoelastic properties of the layers. These variations can be affected by molecular interactions between proteins, nucleic acids and cells, but also by cellular changes such as those affecting the cytoskeleton close to the membrane in contact with the sensing surface. The size of the bacteria can complicate whole-cell analysis due to its large dimensions of 500–2000 nm.

A parameter that is certainly useful for defining the viscoelastic properties of the model under study is the viscosity coefficient *η*. According to the equations of the Kelvin–Voigt model and using the corresponding software we can calculate variations in the viscosity coefficient considering the naked crystal in the absence of any adlayer as the initial state. But since the crystal is initially immersed in water, the value of the initial viscosity coefficient was equal to those of water, 1.0016 mPa·s. Figure 6A shows the trend of the viscosity coefficient changes as a function of time.

Addition of the neutravidin resulted in the increase of the viscosity coefficient by about 1.42 mPa·s. One can assume that because neutravidin forms a compact layer the viscosity increases close to the sensing layer. Furthermore, the value obtained is like those found in the literature regarding protein layers. For example, Dutta et al. [60] studied histones and measured the viscosity changes of these proteins deposited on the QCM electrode. The viscosity coefficient obtained in this work was 1.3–2 mPa·s. Most recently we showed that adsorption of β-casein on the hydrophobic surface formed by dodecanethiol at the QCM crystal is characterized by the viscosity coefficient 0.96 mPa·s (unpublished results).

Following addition of the aptamer, a further increase in the viscosity coefficient was observed to a value 1.81 mPa·s. This agrees well with a previous finding of an increase of motional resistance, *R_m_*, for aptamer layers and agrees well also with results published by Lu et al. [61]. Regarding DNA adsorbed on QCM crystals, they obtained for viscosity coefficient values between 1.24–1.45 mPa·s.

Subsequently, the crystal was incubated with increased concentrations of *Listeria innocua*. It can be seen that the viscosity tended to slightly decrease (Figure 6B). This trend may be related to the fact that the aptamers linked to the bacteria are less free to move in the layer. In fact, studies on the viscosity of DNA in solution (and consequently, more free to move) at different concentrations and in different saline conditions have been carried out, and the lowest value obtained of the viscoelastic coefficient was 3.65 mPa·s [62]. This value is higher than those obtained in our work with the aptamer bound to the neutravidin. It seems that this supports the assumption according to which viscosity could slightly decrease as the mobility of the aptamer decreases. The binding of bacteria with aptamers probably decreases the aptamer mobility and as a result further slight reduction in the viscosity of the nucleic acid, and consequently of the mixed layer is observed. The bacteria would probably play a marginal role in the variation of the total viscosity since, as previously observed, a low number of cells would be bound on the sensor surface. Finally, after the last two incubations, a slight increase in viscosity was observed. This result confirms the theory of coupled resonator formed by bacteria attached on aptamers. Aptamers in this case work as a spring that mediates elastic loading, where an increase in frequency is dependent more on contact stiffness than on deposited mass.

Another important parameter to analyze is the elastic coefficient μ. Figure 7A describes the trend of this parameter during the experiment. As can be seen, the adsorption of neutravidin causes an increase in the elastic coefficient of about 3.6 × 10^5^ Pa. This is a higher value in comparison with those reported in the previously mentioned work by Dutta et al. [60] (1.2–1.6 × 10^4^ Pa), as well as those for β-casein layers: 6.72 × 10^4^ Pa estimated by us (unpublished results). Rather high values of elastic coefficient for neutravidin layers can be due to the formation of a compact structure. Neutravidin has an isoelectric point close to neutral in contrast with negatively charged β-casein or histone proteins, which present a condensation of positive surface charges. Subsequently, the bond of the aptamer causes a decrease in the elastic modulus. This may be because the nucleic acid is viscous and relatively free to move. Thus, the aptamer layer causes a greater dissipation of the energy of the acoustic wave and does not allow the maintenance of the previous elastic condition. The elastic coefficient for the aptamer layer, 2.36 × 10^5^ Pa (a value obtained by calculating the variation with respect to neutravidin and also by considering the variation in elasticity due to the change of liquid medium, from water to PBS), is similar to the *μ* values obtained by Sun et al. (2–4 × 10^5^ Pa) in studies carried out on DNA at different lengths and in different conditions [63]. Similar results were also obtained in other works performed by the same group [58].

The elastic modulus increased slightly following the addition of *Listeria innocua* (Figure 7B). This can be explained by the fact that the bacteria restricted the DNA aptamer motion following binding. Thus, the nucleic acid would decrease the possibility of dissipating the acoustic energy and, therefore, cause a slight increase in the elastic modulus of the multilayer. However, the last two incubations with the highest bacteria concentrations caused a slight decrease in the elastic modulus. This result could again be further confirmation that the system is acting as coupled resonator. In fact, if the aptamers had been removed, we would not have observed a decrease but an increase in the elastic coefficient, tending towards neutravidin values. The last two incubations tended to bring the elastic coefficient back to values present before the incubations with the bacteria, probably because the cells were bound to the aptamer in spring-like manner as discussed in Section 3.2.

## 4. Conclusions

In this study, we showed the high potential of multiharmonic QCM for analysis of the interaction of bacteria *Listeria innocua* with DNA aptamers immobilized at the surface of a QCM transducer covered by neutravidin. By measurements of the changes in resonant frequency, Δ*f*, and dissipation, Δ*D*, and using the Kelvin–Voight model, we estimated also the penetration depth of the acoustics wave, viscosity coefficient and shearing modulus during the formation of sensing aptamer layers and following interaction with bacteria. At a relatively low concentration of *Listeria innocua*, (5 × 10^3^–10^6^ CFU/mL) the decrease of resonant frequency and slight increase of dissipation were observed. This is evidence of interaction of bacteria with the aptamer layer. However, the changes in the frequency were not so high as one would expect considering the size and mass of bacteria (maximal decrease of the frequency at 10^6^ CFU/mL have been 7.37 ± 0.04 Hz). This can be due to the fact that the penetration depth of the acoustic wave (120 nm) is lower in comparison with the thickness of the bacterial layer (500 nm).

The measurement of higher current harmonics and application of the Kelvin–Voight viscoelastic model allowed us to analyze the viscoelastic properties of the sensing layers with adsorbed bacteria. The results obtained suggest substantial variation in viscosity coefficient, *η*, and shearing modulus, *μ*, at various modifications of the QCM transducer. In addition to the mass changes, the effect of the extracellular polymeric substance (EPS) released by bacteria should be considered. The observed increase of resonant frequency and decrease of dissipation at a relatively high concentration of bacteria (>5 × 10^5^ CFU/mL) can be due to the behavior of adsorbed bacterial layer like a spring system. This phenomenon is rather important for optimization of the sensor development and requires further detailed study including application of other physical methods and imaging techniques.

## Figures and Tables

**Figure 1 sensors-21-05585-f001:**
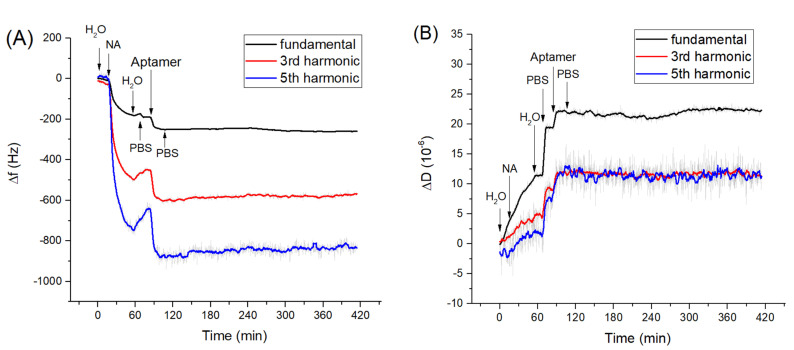
The kinetics of the changes of (**A**) fundamental frequency, 3rd and 5th harmonic frequencies and (**B**) dissipation vs. time following the addition of neutravidin (NA, 125 µg/mL)) and DNA aptamers (0.5 µM). The moment of addition of various compounds as well as washing of the surface by water and phosphate-buffered saline (PBS) are shown by arrows.

**Figure 2 sensors-21-05585-f002:**
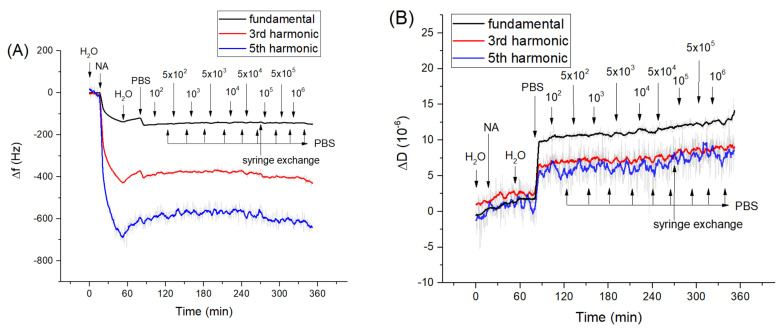
The kinetics of the changes of (**A**) fundamental frequency, 3rd and 5th harmonic frequencies and (**B**) dissipation vs. time following addition of *Listeria innocua* to the neutravidin layer. The moment of addition of neutravidin (NA), bacteria and washing of the surface by deionized water, PBS as well as exchange of the syringe from the syringe pump are shown by arrows. The syringe exchange caused a slight increase of dissipation at a fundamental harmonic.

**Figure 3 sensors-21-05585-f003:**
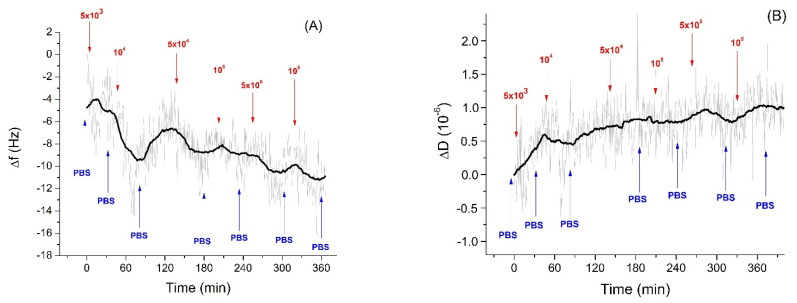
The kinetics of the changes of (**A**) fundamental frequency and (**B**) dissipation of the quartz crystal microbalance (QCM) transducer covered by neutravidin and aptamers following addition of *Listeria innocua* in various concentrations (in CFU/mL). The moment of addition of bacteria as well as washing of the surface by PBS are shown by arrows. Figure 3A was reproduced from [17].

**Figure 4 sensors-21-05585-f004:**
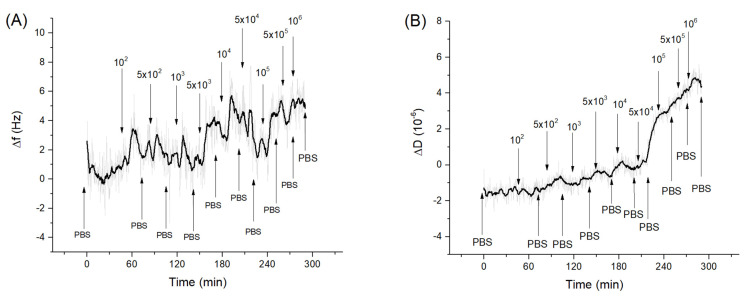
The kinetics of the changes of (**A**) fundamental frequency and (**B**) dissipation of the QCM transducer covered by neutravidin and aptamers specific to *Listeria* spp. following addition of *E. coli* in various concentrations (in CFU/mL). The moment of addition of bacteria as well as washing of the surface by PBS are shown by arrows.

**Figure 5 sensors-21-05585-f005:**
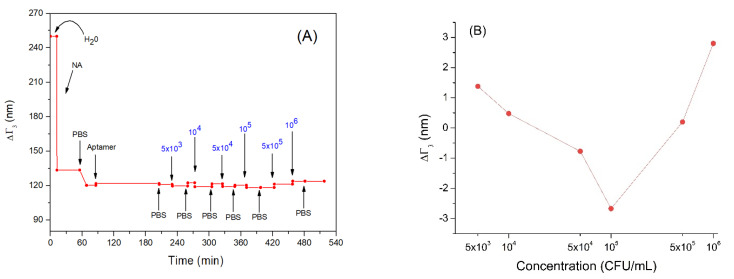
(**A**) the kinetics of the changes of penetration depth during formation of the aptamer layer and addition of *Listeria innocua* (in CFU/mL). (**B**) Changes in penetration depth depending on applied bacteria concentration in logarithmic scale. Addition of various compounds and washing the surface by water or PBS is shown by arrows.

**Figure 6 sensors-21-05585-f006:**
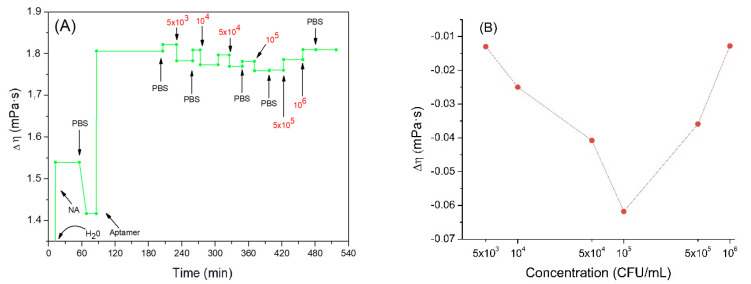
(**A**) the kinetics of the changes of viscosity coefficient (the value of the water viscosity coefficient is not shown in the figure to better highlight the subsequent changes) during formation of the aptamer layer and following addition of *Listeria innocua* (in CFU/mL). (**B**) Changes in viscosity coefficient depending on applied bacteria concentration in logarithmic scale. Addition of various compounds and washing the surface by water or PBS is shown by arrows.

**Figure 7 sensors-21-05585-f007:**
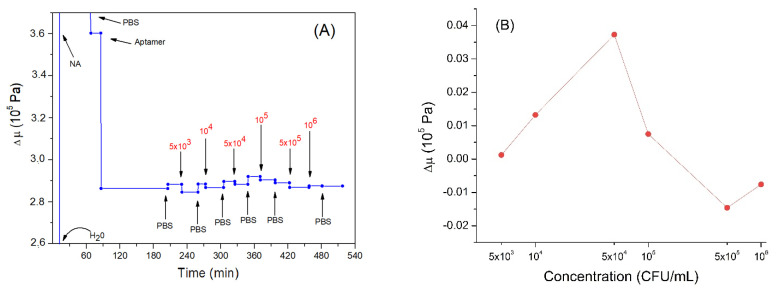
(**A**) the kinetics of the elastic coefficient during formation of the aptamer layer and following addition of *Listeria innocua* (in CFU/mL). The first three values are not shown in the graph, to better highlight the subsequent variations. (**B**) changes in elastic coefficient depending on applied bacteria concentration in logarithmic scale. Addition of various compounds and washing the surface by water or PBS is shown by arrows.

## Data Availability

Not applicable.

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
