# Peer review of "Changes of Viscoelastic Properties of Aptamer-Based Sensing Layers Following Interaction with Listeria innocua"

_sensors, 2021, doi:10.3390/s21165585_

Round 1

Reviewer 1 Report

The manuscript describes the development of an acoustic aptasensor for the detection of foodborne bacteria Listeria innocua. Part of the results concerning the analytical performances of the aptasensor were reported in a previous paper and the main novelty here consists in the investigation of the contributions to the sensor’s response at different concentrations of bacteria, related to the changes in dissipation and in the viscoelastic properties of the aptamer-based sensing layer . While such analysis is definitely important, the information presented appears as a half-said story: the authors advanced a hypothesis to explain their experimental data, provided some arguments but presented no additional investigation by parallel methods to verify their hypothesis. Revisions are necessary before recommending for publication in Sensors.

Specific comments:

- The authors should carefully check and either eliminate the already presented data, such as Figure 1A which was already presented in the Proceedings paper (reference 21). Alternatively, if the respective data/graph is really needed for the discussion, it should be mentioned that is reproduced from reference 21.

-Spelling errors need to be corrected thoughout the manuscript. Line 18 : “senor”, line 258 “has elaborates”; line 266 “method absence/presence”; line 323 “increa”; line443: “does not allowing” etc

-Introduction lines 134-136: what are the O and H antigens and what is their relevance for the aptasensor used in this work?

-Section 2.3 Preparation of the aptasensor: was the adsorption of neutravidin enough to ensure the uniformous and stable coverage of the crystal? How was the non-specific adsorption checked and prevented? The variation of the baseline in the presence of PBS only during the 360 min experiment should be shown in parallel to demonstrate the stability of the sensing layer and the lack of drift.  Was the temperature controlled/constant during the experiment?

-Lines 259-261” what means “ the highest possible probability in accordance to the frequency and dissipation variation inserted in the model”?

-Lines 276-278: why was neutravidin dissolved in distilled water and not in a buffer (since is a protein)?

-The noise seems pretty high ( Figure 2 A and B). Statistical evidence is needed to show that the changes induced following the addition of bacteria solutions of different concentrations are relevant.

-Why was a solution of 0.5 µM aptamer used to obtain the aptasensor, was this concentration optimized?

-Since the dissipation data presented in Figure 1B indicate significant disssipation ( ΔD = (5.1±1.1)x10-6) following the aptamer attachment, and consequently “ the Sauerbrey equation is not strongly valid for aptamer adlayer” (line 317), it is OK to continue with the quantitative measurements of bacteria?

-Since the application of Sauebrey equation is not entirely valid and the chance in the resonant frequency of the aptasensor provides only an estimation of the aptamer coverage, the authors should try to confirm this value of aptamer coverage by a parallel method, e.g. SPR or electrochemistry.

Lines 348-352: the observation of the change in the resonant frequency of the sensor at high bacterial load occurring “occasionally during application of the higher concentration of bacteria (above 106 CFU/mL)”, “usually with older samples” is not supported by any presented data in the manuscript. It should be either removed or revised to include supporting information.

Lines 354-361: the specificity refers to data already reported in reference 21. Control experiments with the sensor modified with neutravidin but without aptamer or with a different aptamer should be included to demonstrate that the aptasensor’s response is really due to the specific binding of the Listeria innocua by the aptamer at the sensor surface.

-Lines 388-389: “This could be explained by the fact that the sensor is losing material, such as previously deposited bacteria and/or aptamer”. Considering the strong biotin-neutravidin bond, the hypothesis of aptamer removal would imply that neutravidin will be lost as well. The advanced hypothesis should have been checked by measuring the coverage with aptamer/neutravidin by an independent method (e.g. electrochemical etc)   

-Lines 426-430: the value of the viscosity coefficient change following the addition of the aptamer should be mentioned in the text to support the idea of similar value with those reported in literature.

-Lines 443-446: if the sensor has lost some material during the incubation with 5x105 and 1x106 cfu/mL bacteria, wouldn’t this be seen as an increase in the resonant frequency as well in Figure 2A?

-The authors should clarity what is the information derived from the use of 3rd and 7th harmonic, in addition to fundamental frequency in Figure 1? Can this be used to support/ invalidate the application of the Sauerbrey equation based on similar or different Δf/n at the fundamental, 3rd and 7th harmonic after the sensor’s modification with neutravidin and aptamer, respectively?

-Figure 1A: the label of the Y axis should be revised; according to the figure caption it should be “Δf/n”  where n is the harmonics number

-The authors have used in this work the A15 aptamer reported in reference 30 (Duan et al, 2013) which has significantly higher affinity for Listeria monocytogenes than for Listeria innocua, other Listeria strains, Staphylococcus aureus, thyphimurium and E coli. (Why have the authirs focused on Listeria innocua and not on Listeria monocytogenes?).

From reference 30 (Duan et al, 2013) the response for Listeria innocua is only about 25-30% higher than for E Coli at a 108 bacterial concentration. What is the explanation for the significant differences observed with the QCM biosensor, reported in authors’ previous work (reference 21), i.e., increase in resonance frequency observed for E coli versus decrease in frequency for the same concentration of Listeria innocua? Are these due to (1) particularities of the immobilized aptamer, (2) the detection method (QCM) enabling to observe finer details of the aptamer-ligand binding than the fluorescence method used in reference 30, (3) the high concentrations of bacteria used in the specificity assay in reference 30 or (4) another explanation? The aptasensor’s specificity deserves further analysis in the manuscript.

Author Response

Comment: The manuscript describes the development of an acoustic aptasensor for the detection of foodborne bacteria Listeria innocua. Part of the results concerning the analytical performances of the aptasensor were reported in a previous paper and the main novelty here consists in the investigation of the contributions to the sensor’s response at different concentrations of bacteria, related to the changes in dissipation and in the viscoelastic properties of the aptamer-based sensing layer . While such analysis is definitely important, the information presented appears as a half-said story: the authors advanced a hypothesis to explain their experimental data, provided some arguments but presented no additional investigation by parallel methods to verify their hypothesis. Revisions are necessary before recommending for publication in Sensors.

Response: We are grateful to the reviewer for most useful comments and suggestions that allowed us to improve the manuscript. In revised manuscript we addressed all reviewer's comments. The detailed response is presented below. All additions and changes made in the revised manuscript are highlighted by yellow. We should also point out that the title of the paper has been changed in order to underline the focus of the manuscript on the analysis of the viscoelastic properties of the sensing layers. Additional discussion and explanation were made based also on QCM-D investigations of other authors focused on the study of interactions of bacteria with various surfaces.

Specific comments:

Comment: The authors should carefully check and either eliminate the already presented data, such as Figure 1A which was already presented in the Proceedings paper (reference 21). Alternatively, if the respective data/graph is really needed for the discussion, it should be mentioned that is reproduced from reference 21.

Response: Figure 1A is necessary for understanding of the results. We mentioned that it is reproduced from reference [17] as recommended. (Due to rearrangement and shortening of the Introduction and inclusion additional references the numbering of references was changed).

Comment: Spelling errors need to be corrected throughout the manuscript. Line 18 : “senor”, line 258 “has elaborates”; line 266 “method absence/presence”; line 323 “increa”; line443: “does not allowing” etc

Response: Thank you for careful reading of the manuscript. All spelling errors were corrected.

Comment: Introduction lines 134-136: what are the O and H antigens and what is their relevance for the aptasensor used in this work?

Response: E. coli O157:H7 expresses somatic antigen O157 and flagella antigen H7. This has been clarified on page 4, lines 153-154 of revised manuscript. Binding of this serotype by its specific aptamer is aim of several research groups and their publications. However, in our work we applied only aptamer sensitive to Listeria. E. Coli, strain B40 has been used for clarification of specificity of Listeria based aptamer. In revised manuscript more information about strains used was provided in page 4, lines 155-174.

Comment: Section 2.3 Preparation of the aptasensor: was the adsorption of neutravidin enough to ensure the uniformous and stable coverage of the crystal? How was the non-specific adsorption checked and prevented? The variation of the baseline in the presence of PBS only during the 360 min experiment should be shown in parallel to demonstrate the stability of the sensing layer and the lack of drift.  Was the temperature controlled/constant during the experiment?

Response: Neutravidin formed dense and stable layer on the crystal surface, which should provide sufficient coverage for aptamer adsorption and prevent nonspecific adsorption by bacteria as it has been estimated based on the frequency changes. This was partially approved in experiment with E. Coli, where no decrease of the resonant frequency was observed. Stability of the sensing layer was checked before application of the bacteria for 1-2 hours and we evaluated the possibility of drift for completed sensing layer. As it can be seen from Figure 1A, immobilization of aptamers at the neutravidin surface provides stable baseline with minimal drift. Experiments were performed in laboratory conditions and temperature was controlled with precision of 0.5 oC We included this information at page 8, line 345 of revised manuscript.

Comment: Lines 259-261” what means “ the highest possible probability in accordance to the frequency and dissipation variation inserted in the model”?

Response: Thank you for this comment. The term “the highest possible probability” is certainly not relevant and we removed it from the revised manuscript.

Comment: -Lines 276-278: why was neutravidin dissolved in distilled water and not in a buffer (since is a protein)?

Response: According to the producer (Thermo Scientific, https://assets.thermofisher.com/TFS-Assets/LSG/manuals/MAN0011245_NeutrAvidin_Biotin_BindProtein_UG.pdf) it is recommended to dilute neutravidin in ultrapure water, because it is not directly soluble in PBS. We added this information in revised manuscript at page 7, lines 302-304.

Comment: The noise seems pretty high (Figure 2 A and B). Statistical evidence is needed to show that the changes induced following the addition of bacteria solutions of different concentrations are relevant.

Response: Each frequency and dissipation values were extracted from at least 3 independent experiments and the standard deviation has been determined.

Comment: -Why was a solution of 0.5 µM aptamer used to obtain the aptasensor, was this concentration optimized?

Response: Using experience from our previous research, we have found that formation of the aptamer layer requires typically aptamer concentration in the range 0.5 µM -1 µM to provide sufficient coverage for the QCM surface covered by neutravidin. We also estimated that at the aptamer concentration of 0.5 μM all available binding sites of neutravidin are covered by aptamers.

Comment: -Since the dissipation data presented in Figure 1B indicate significant dissipation (ΔD = (5.1±1.1)x10-6) following the aptamer attachment, and consequently “ the Sauerbrey equation is not strongly valid for aptamer adlayer” (line 317), it is OK to continue with the quantitative measurements of bacteria?

Response: Thank you for this important comment. Sauerbrey equation is certainly not valid for evaluation of the changes of the mass caused by adsorption of the bacteria. As it has been explained in revised manuscript the changes in resonant frequency are only partially caused by mass of the bacteria due to the fact that decay length of the sensing layer (120 nm) is lower in comparison with the thickness of bacteria (500 nm). In addition, adsorption of bacteria to the surface can be accompanied also by release of the extracellular polymeric substance (EPS), which can cause non-specific interactions of bacteria with aptamer layers. This phenomenon requires special study. However, for practical purposes the changes in the fundamental frequency can be used for qualitative examination of the binding of bacteria to the aptamer layer. However, measurement of both frequency changes and dissipation allowed us to analyze the changes in viscoelastic properties of the sensing layers also following adsorption of bacteria. We included additional explanation at pages 8-9, lines 356-372 of the revised manuscript.

Comment: Since the application of Sauebrey equation is not entirely valid and the chance in the resonant frequency of the aptasensor provides only an estimation of the aptamer coverage, the authors should try to confirm this value of aptamer coverage by a parallel method, e.g. SPR or electrochemistry.

Response: We indeed are planning to use parallel methods in the future experiments. However, based also on the comments of the reviewer we changed the focus of the manuscript on the analysis of viscoelastic properties of the aptamer-based sensing layers and changed correspondingly also the Title, Abstract and Conclusion.

Comment: Lines 348-352: the observation of the change in the resonant frequency of the sensor at high bacterial load occurring “occasionally during application of the higher concentration of bacteria (above 106 CFU/mL)”, “usually with older samples” is not supported by any presented data in the manuscript. It should be either removed or revised to include supporting information.

Response: The phenomenon concerning increase of the resonant frequency has been additionally discussed also in terms of the results of other authors that reported similar observation for adsorption of the bacteria on protein layers (pages 8-9, lines 354-370 of revised manuscript). Corresponding figure has been included in Supplementary data as Figure S1.

Comment: Lines 354-361: the specificity refers to data already reported in reference 21. Control experiments with the sensor modified with neutravidin but without aptamer or with a different aptamer should be included to demonstrate that the aptasensor’s response is really due to the specific binding of the Listeria innocua by the aptamer at the sensor surface.

Response: As we already mentioned above, due to tight aptamer array, it is no space for adsorption of bacteria to the neutravidin layer. Also experiments with E. coli evidence that adsorption of this bacteria resulted in increase of resonant frequency and not in its decrease as it was in the case of specific interaction of Listeria innocua. We should also mention that in recent work we confirmed that breast cancer cells do not interact with neutravidin layer (Poturnayova et al. Biosensors, 2019, 9, 72; doi:10.3390/bios9020072). However, we agree with the reviewer that such experiments should be performed. But this is very specific investigation. Based on the studies of other authors it can be expected that certain non-specific interactions of bacteria with neutravidin layers could be expected considering possible effect of extracellular polymeric substance (EPS). We are planning to perform such experiments in a near future.

Comment: Lines 388-389: “This could be explained by the fact that the sensor is losing material, such as previously deposited bacteria and/or aptamer”. Considering the strong biotin-neutravidin bond, the hypothesis of aptamer removal would imply that neutravidin will be lost as well. The advanced hypothesis should have been checked by measuring the coverage with aptamer/neutravidin by an independent method (e.g. electrochemical etc)

Response: By analyzing of the available data of other authors that studied the interaction of bacteria with protein layers, we changed our opinion on the possible lost of the material from the QCM surface. We included following explanation of this phenomena at pages 8-9, lines 356-372 of the revised manuscript and added the possible scheme of interaction of bacteria with sensing layers (Figure S2 of Supplementary data).

We agree with reviewer, that further analysis is required. However, this can be topic of separate article. We are planning to do this in a near future.

Comment: Lines 426-430: the value of the viscosity coefficient change following the addition of the aptamer should be mentioned in the text to support the idea of similar value with those reported in literature.

Response: We added the value of viscoelastic coefficient from our measurement as recommended (page 12, lines 493-494).

Comment: Lines 443-446: if the sensor has lost some material during the incubation with 5x105 and 1x106 cfu/mL bacteria, wouldn’t this be seen as an increase in the resonant frequency as well in Figure 2A?

Response: This effect was observed only once, but we were concerned about origin of this phenomenon. We have noticed that it happened during application of earlier-prepared sample, so we assumed bacteria were still reproducing and real concentration in the sample is much higher. We decided to take this event as opportunity to present multi-frequency QCM capabilities to solve previously unknown reasons of mass deposition. We included the figure where the increase of the frequency is visible at higher concentrations of bacteria as Figure 1S of Supplementary data.

Comment: -The authors should clarity what is the information derived from the use of 3rd and 7th harmonic, in addition to fundamental frequency in Figure 1? Can this be used to support/ invalidate the application of the Sauerbrey equation based on similar or different Δf/n at the fundamental, 3rd and 7th harmonic after the sensor’s modification with neutravidin and aptamer, respectively?

Response: For QCM-D experiment it is important to estimate the penetration depth. The acoustics wave of the fundamental frequency penetrates deeper into the liquid, while higher harmonics penetrate only into to the densely packed layers. Therefore, fundamental frequency is useful for construction calibration curve, while higher harmonics are suitable for analysis of viscoelastic properties of the sensing layers. The following text has been added at page 10, lines 431-436.

"For this purpose the data on the changes of frequency and dissipation for fundamental and up to 7th harmonics (3rd, 5th and 7th) were crucial. While the acoustics wave of the fundamental frequency penetrates deeper into the liquid, the higher harmonics penetrate only into to the densely packed layers. Therefore, fundamental frequency is useful for the construction of calibration curve, while higher harmonics are suitable for analysis of viscoelastic properties of the sensing layers [32]."

Comment: -Figure 1A: the label of the Y axis should be revised; according to the figure caption it should be “Δf/n”  where n is the harmonics number

Response: We changed the label of Y axis as recommended. Thank you for your correction.

Comment: The authors have used in this work the A15 aptamer reported in reference 30 (Duan et al, 2013) which has significantly higher affinity for Listeria monocytogenes than for Listeria innocua, other Listeria strains, Staphylococcus aureus, thyphimurium and E coli. (Why have the authors focused on Listeria innocua and not on Listeria monocytogenes?).

Response: A15 aptamer has indeed higher affinity for Listeria monocytogenes. However, due the risk on working with very infectious Listeria monocytogenes, we decide to work only with less infective Listeria innocua, that also interact with aptamer compared to other bacteria in above mentioned article. We added corresponding explanation in the revised manuscript at page 3, lines 131-136.

In addition, we provided more detailed information about bacterial strands used at page 4.

Comment: From reference 30 (Duan et al, 2013) the response for Listeria innocua is only about 25-30% higher than for E Coli at a 108 bacterial concentration. What is the explanation for the significant differences observed with the QCM biosensor, reported in authors’ previous work (reference 21), i.e., increase in resonance frequency observed for E coli versus decrease in frequency for the same concentration of Listeria innocua? Are these due to (1) particularities of the immobilized aptamer, (2) the detection method (QCM) enabling to observe finer details of the aptamer-ligand binding than the fluorescence method used in reference 30, (3) the high concentrations of bacteria used in the specificity assay in reference 30 or (4) another explanation? The aptasensor’s specificity deserves further analysis in the manuscript.

Response: Because validation of detection capabilities of the bacterial biosensor requires more measurements, we decided to focus of the manuscript on the analysis of viscoelastic properties of the immobilized aptamer layers and those with bounded bacteria. However, we agree with reviewer suggestions and included following text in the revised manuscript at page 10, lines 406-424.

„Application of these concentrations of E. coli caused only slight increase of fundamental frequency and practically does not change the dissipation. However, substantial increase of frequency took place for 3rd and even more for 7th harmonics. This increase was accompanied by decrease of dissipation (Figure S3 of Supplementary data). The insignificant changes of fundamental frequency are evidence of non-specific interaction of E. coli with aptamer surface. However, increase of the frequency of higher overtones suggest non-specific interaction of bacteria with the sensing layer. Bacterial adsorption can be considered as a coupled oscillators. Depending on the stiffness of the spring connected the bacteria with the sensing surface, the frequency shift can be more or less positive (See Figure S2 of Supplementary data for explanation). In comparison, Listeria innocua caused significant frequency decrease already at 5×103 CFU/mL. This result is surprising considering that the aptamer used has only slightly less affinity to E. coli in comparison with Listeria innocua [35] as determined by fluorescence method. At the same time the interaction of bacteria with the surfaces is promoted also by release of extracellular polymeric substance (EPS), which can cause non-specific interactions. However, as it has been shown by Olsson et al. [54], EPS is more strongly connected with the QCM substrate and increases the mass loading. Therefore, EPS can contribute to the negative frequency shift. At higher concentrations of bacteria, the effect of elastic-spring coupling described above can promote the frequency increase.

We also included Figure S3 in Supplementary data on the changes of frequency and dissipation in multiharmonic measurements following addition of E. Coli. These data evidence on specificity of the sensor for Listeria innocua.

Reviewer 2 Report

In this work, the authors described an interesting multi-harmonic QCM biosensor for Listeria innocua detection. The whole detection protocol is quick and sensitive. The obtained results suggest that the aptamer based QCM sensor is specific 494 and sufficiently rapid for pathogen detection with relatively low LOD. This work meets the aim and scope of the journal, and I would recommend publication with some minor revision.

  1. The authors demonstrated a novel multi-harmonic QCM protocol. Please specify the advantages of this design comparing with traditional biosensors.
  2. Section 2.1. Please explain DNA aptamer used in this work. Why this DNA sequence is selected in this work? Is it specific for Listeria innocua only?
  3. Section 3, line 264. Why coli is also determined in this work since it is for Listeria innocua detection?
  4. It seems the results were mainly detected based on the viscoelastic parameters. However, the authors demonstrated this is a harmonic design. The authors should explain how harmonic parameters were monitored and decided for Listeria innocua
  5. The authors demonstrated LOD for Listeria innocua detection is determined by 3 SD/k, and 1.6 * 103 CFU/mL was obtained. It seems this method could meet the requirement with 103 CFU/mL concentration in food derived illness. The authors should compare this developed protocol with publications for Listeria innocua
  6. The authors described and validated this multi-harmonic QCM protocol for Listeria innocua detection with DNA aptamers. As a newly developed methods, accuracy and precision and detection range should also be validated.
  7. Listeria innocua is a food derived pathogen. The authors should apply this developed protocol for commercial samples to identify the viability of this method.

Author Response

Comment: In this work, the authors described an interesting multi-harmonic QCM biosensor for Listeria innocua detection. The whole detection protocol is quick and sensitive. The obtained results suggest that the aptamer based QCM sensor is specific and sufficiently rapid for pathogen detection with relatively low LOD. This work meets the aim and scope of the journal, and I would recommend publication with some minor revision.

Response: We are grateful to the reviewer for positive opinion and for most useful comments and suggestions that allowed us to improve the manuscript. We addressed all reviewer's comments. The detailed response is presented below. All additions and changes made in the revised manuscript are highlighted by yellow.

Comment: The authors demonstrated a novel multi-harmonic QCM protocol. Please specify the advantages of this design comparing with traditional biosensors.

Response: Traditional methods for bacteria detection allow determination of bacteria with sufficient sensitivity and specificity. However, these methods require well equipped specialized laboratories and trained staff. The approach proposed in our work allowing simple sensor fabrication and relatively fast detection of bacteria. Additional advantage of using QCM method with dissipation consists in possibility of evaluation of viscoelastic properties of the sensing layers. This can be also important for the study of adhesion of bacteria on various surfaces. In the revised manuscript we explained in more details the advantage of multiharmonic analysis.

Comment: Section 2.1. Please explain DNA aptamer used in this work. Why this DNA sequence is selected in this work? Is it specific for Listeria innocua only?

Response: A15 aptamer used in our work has higher affinity for Listeria monocytogenes. However, due the risk on working with very infectious Listeria monocytogenes, we decide to work only with less infective Listeria innocua, that also interact with aptamer compared to other bacteria in above mentioned article. We added following explanation in the revised manuscript at page 3, lines 130-135, as follows:

This aptamer (A15) has been developed for detection of bacterial pathogen Listeria monocytogenes [35]. However, according to the measurement of fluorescence intensity this aptamer modified at 5' end by fluorescein can bind, although with less affinity, the Listeria innocua. The later is much less infectious in comparison with Listeria monocytogenes. Therefore, this bacteria has been used in our study.

In addition, we provided more detailed information about bacterial strands used at page 4.

Comment: Section 3, line 264. Why coli is also determined in this work since it is for Listeria innocua detection?

Response: E. coli served as the alternative bacteria for analysis of the specificity of detection, as the aptamer used is less specific for E. coli in comparison with those of Listeria innocua.

Comment: It seems the results were mainly detected based on the viscoelastic parameters. However, the authors demonstrated this is a harmonic design. The authors should explain how harmonic parameters were monitored and decided for Listeria innocua

Response: The higher current harmonics have been monitored thanks to the network analyzer that is capable to perform such an analysis. In the revised manuscript we also included explanation of the significance of higher harmonic analysis in the evaluation of viscoelastic properties of sensing layers. The following text has been added at page 10, lines 431-436.

"For this purpose the data on the changes of frequency and dissipation for fundamental and up to 7th harmonics (3rd, 5th and 7th) were crucial. While the acoustics wave of the fundamental frequency penetrates deeper into the liquid, the higher harmonics penetrate only into to the densely packed layers. Therefore, fundamental frequency is useful for the construction of calibration curve, while higher harmonics are suitable for analysis of viscoelastic properties of the sensing layers [32].“

Comment: The authors demonstrated LOD for Listeria innocua detection is determined by 3 SD/k, and 1.6 * 103 CFU/mL was obtained. It seems this method could meet the requirement with 103 CFU/mL concentration in food derived illness. The authors should compare this developed protocol with publications for Listeria innocua.

Response: There are few publications that are dedicated to detection of Listeria innocua. We achieved better LOD than mentioned publication (104 CFU/mL by Uusitalo et al., 2016 [58] using SERS method). The following text has been added into the revised manuscript, pages 9-10, lines 398-400:

The LOD obtained by QCM method is better in comparison with SERS based immunosensor [58] for which the LOD of 104 CFU/mL for L. innocua has been determined.

Comment: The authors described and validated this multi-harmonic QCM protocol for Listeria innocua detection with DNA aptamers. As a newly developed methods, accuracy and precision and detection range should also be validated.

Response: We decided to focus our article on the analysis of viscoelastic properties of bacteria bound on aptamer layer. Therefore, we changed the Title of the article, Abstract and Conclusion and performed substantial modifications of the text. Further detailed research is planned in a near future focused on the comparative analysis of the interaction with the sensing surface of the Listeria innocua and Listeria monocytogenes.

Comment: Listeria innocua is a food derived pathogen. The authors should apply this developed protocol for commercial samples to identify the viability of this method.

Response: We found it crucial to test QCM biosensor on the isolated bacteria sample from contaminated food products, as it is standard method for other techniques. We furthermore planning the testing of the sensor after further optimization.

Reviewer 3 Report

This is an interesting research work with potential application in detection of harmful bacteria in food. The manuscript could be published after addressing the following problems:

  1. The process of binding of Listeria innocua to the aptamer was not described in detail.

According to the ref. [30] the aptamer was designed for Listeria monocytogenes which may have several binding sites (e.g. lipopolysaccharide, membrane proteins, lipoproteins). The reported in Ref. [30] binding affinity is high with KD of about 48 nM. It is a question, however, whether the selected aptamer is suitable for detection of Listeria innocua.

  1. The observed changes in frequency and dissipation (Fig. 2) upon binding Listeria innocua are suspiciously small, particularly considering the large mass of the target analyte. Binding of much smaller molecules such as neutravidin and aptamer produced substantially larger changes. The explanation given by authors that the acoustic wave does not penetrate deep into the adsorbed molecular layer is not correct, since QCM sensor gives the response to the  total added mass. Maybe, the analyte (bacteria) is two heavy, and thus beyond the limit of Sauerbrey equation.  In that case, however, the dissipation could be very high. It is possible to observe this by plotting the frequency dependence of the impedance.  
  2. An alternative explanation is that the aptamer is not too specific to Listeria innocua. This could be checked by comparing the dissociation constant KD with that given in ref. [30]. The value of KD of 105 CFU is not helpful; KD should be given in molar units.
  3. In practical terms, the sensor having responses comparable with the noise (see Fig. 2) is not useful. I don’t agree with the evaluation of LOD; the triple noise level is much higher (about 10 Hz) which gives LOD above 106 CFU the maximal concentration used.

Author Response

Comment: This is an interesting research work with potential application in detection of harmful bacteria in food. The manuscript could be published after addressing the following problems.

Response: We are grateful to the reviewer for positive opinion and for most useful comments and suggestions that allowed us to improve the manuscript. We addressed all reviewer's comments. The detailed response is presented below. All additions and changes made in the revised manuscript are highlighted by yellow.

Comment: The process of binding of Listeria innocua to the aptamer was not described in detail. According to the ref. [30] the aptamer was designed for Listeria monocytogenes which may have several binding sites (e.g. lipopolysaccharide, membrane proteins, lipoproteins). The reported in Ref. [30] binding affinity is high with KD of about 48 nM. It is a question, however, whether the selected aptamer is suitable for detection of Listeria innocua.

Response: Aptamer from ref. [30] showed slightly higher specificity towards Listeria innocua than other bacteria except Listeria monocytogenes, for which it is most specific. We added corresponding explanation in the revised manuscript at page 3, lines 130-135, as follows:

“This aptamer (A15) has been developed for detection of bacterial pathogen Listeria monocytogenes [35]. However, according to the measurement of fluorescence intensity this aptamer modified at 5' end by fluorescein can bind, although with less affinity, the Listeria innocua. The later is much less infectious in comparison with Listeria monocytogenes. Therefore, this bacteria has been used in our study.“

In addition, we provided more detailed information about bacterial strands used at page 4.

Furthermore, we focused our article more on analysis of viscoelastic properties. Therefore, we changed the Title, Abstract, Conclusion and clarified this approach in the revised text.

Comment: The observed changes in frequency and dissipation (Fig. 2) upon binding Listeria innocua are suspiciously small, particularly considering the large mass of the target analyte. Binding of much smaller molecules such as neutravidin and aptamer produced substantially larger changes. The explanation given by authors that the acoustic wave does not penetrate deep into the adsorbed molecular layer is not correct, since QCM sensor gives the response to the  total added mass. Maybe, the analyte (bacteria) is two heavy, and thus beyond the limit of Sauerbrey equation.  In that case, however, the dissipation could be very high. It is possible to observe this by plotting the frequency dependence of the impedance.

Response: Adhesion of the bacteria cell on QCM surface was often described as “coupled oscillator” model. We added this explanation in the revised text. This explains how even the heavy load of the bacteria is influenced by flow and how the dissipation values help to prevent wrong calculation of the mass.

Comment: An alternative explanation is that the aptamer is not too specific to Listeria innocua. This could be checked by comparing the dissociation constant KD with that given in ref. [30]. The value of KD of 105 CFU is not helpful; KD should be given in molar units.

Response: Aptamer is indeed less specific to Listeria innocua then to Listeria monocytogenes. Therefore, we dedicated this article more to the analysis of viscoelastic properties and how the cell adhesion influences the characteristics of the sensing layers. We agree with the comment concerning expression of KD in molar units. However, KD value has been determined in the manuscript was based on the Langmuir isotherm from the frequency changes vs. concentration of bacteria, that were expressed in CFU/mL. In fact, this dissociation constant reflects the stability of the bacteria-sensing surface complexes.

Comment: In practical terms, the sensor having responses comparable with the noise (see Fig. 2) is not useful. I don’t agree with the evaluation of LOD; the triple noise level is much higher (about 10 Hz) which gives LOD above 106 CFU the maximal concentration used.

Response: We did not use measurement noise for LOD determination, but SD was calculated by using 3 frequency change values from 3 independent applications for each concentration.

Reviewer 4 Report

  • Authors introduced acoustic sensor for pathogenic microorganism like Listeria using DNA aptamers. Since the QCM is very helpful method for biosensing in many ways and using aptamer has somewhat novelty in detection of bacteria. However, there are some issues which should be revised to be more appropriate for publication as follows:

(1) Authors need to provide "dry" QCM measurement data if possible. The acoustic wave in aqueous phase has somewhat deviations depending on even density of aqueous phase (rho "el") as authors described in equation (3). Actually, the introduction of many different concentrations of Listeria, e.g. 10~1000 CFU/mL, would be resulted in different density.

(2) For the LOD of sensor, authors are requested to present "optical density of the CFU/mL" for example in 600 nm absorbance, to provide more helpful detection range of cell density.

(3) Authors need to provide pros and cons about using the 35 mer DNA aptamer. There should be long DNA aptamer for Listeria. It also influences the visco-elasticity or elasticity of layer.  

(4) How many times were effective or meaningful measurement for multiple detection ? It is well known that QCM has limitation for multiple bindings on gold surfaces of the QCM. Since the changes of CFU/mL were tried several times, there should be a certain conclusion for re-usability issue,

(5)  The introduction part is somewhat long and tedious in page 2. The introduction part should be more focused on recent QCM itself and aptamer probe for QCM.

(6) There are grammar errors and typos; in page 1, line 38~39; page 6, line 283~284, etc.  

Author Response

Comment: Authors introduced acoustic sensor for pathogenic microorganism like Listeria using DNA aptamers. Since the QCM is very helpful method for biosensing in many ways and using aptamer has somewhat novelty in detection of bacteria. However, there are some issues which should be revised to be more appropriate for publication as follows:

Response: We are grateful to the reviewer for positive opinion and for most useful comments and suggestions that allowed us to improve the manuscript. We addressed all reviewer's comments. The detailed response is presented below. All additions and changes made in the revised manuscript are highlighted by yellow.

Comment: (1) Authors need to provide "dry" QCM measurement data if possible. The acoustic wave in aqueous phase has somewhat deviations depending on even density of aqueous phase (rho "el") as authors described in equation (3). Actually, the introduction of many different concentrations of Listeria, e.g. 10~1000 CFU/mL, would be resulted in different density.

Response: The QCM method is certainly suitable for measuring dry crystals. However, we changed the focus of the article more for evaluation of viscoelastic properties of the sensing layers that are in contact with aqueous media. For this purpose, the measurements were performed in a liquid. At dry conditions it is impossible to provide suitable conformation of the aptamers as well as living bacterial cells.

Comment: (2) For the LOD of sensor, authors are requested to present "optical density of the CFU/mL" for example in 600 nm absorbance, to provide more helpful detection range of cell density.

Response: Missing OD value (0.1 OD at 600 nm) was added into the revised text (page 4, line 185).

Comment: (3) Authors need to provide pros and cons about using the 35 mer DNA aptamer. There should be long DNA aptamer for Listeria. It also influences the visco-elasticity or elasticity of layer.

Response: To provide the most efficient detection of bacteria by QCM aptasensor and its viscoelastic properties, we should arrange the thinnest layer possible for adsorption. Because even short aptamer (several nm) provided effective interaction with bacteria, as stated in our and other articles, we choose it for biosensor development. We also plan to use aptamers with various supporting part  to compare the results with 35 mer aptamer. 

Comment: (4) How many times were effective or meaningful measurement for multiple detection? It is well known that QCM has limitation for multiple bindings on gold surfaces of the QCM. Since the changes of CFU/mL were tried several times, there should be a certain conclusion for re-usability issue,

Response: As there was always a decrease in resonance frequency after the Listeria innocua application, we found “titration” method suitable for analysis. We plan to test possibility of aptamer layer regeneration by various reagents, such as glycine, NaOH, concentrated NaCl. However, this was out of the main concept of the article, which is focused mainly on the analysis of viscoelastic properties of the sensing layers.

Comment: (5) The introduction part is somewhat long and tedious in page 2. The introduction part should be more focused on recent QCM itself and aptamer probe for QCM.

Response: The Introduction has been shortened and rewrite as recommended by reviewer.

Comment: (6) There are grammar errors and typos; in page 1, line 38~39; page 6, line 283~284, etc.

Response: Grammar errors and typos were corrected and whole article was checked several times for even more. Thank you for you time and all the corrections.

Round 2

Reviewer 1 Report

The quality of the manuscript has significantly improved after the revision however there are some important points still not addressed and some remaining concerns. The main concerns are: 1) there is still too much overlap with authors’ previous work reported in reference 17. 2).The noise seems very high in most experiments. Since the raw signals are very small, and there is no convincing proof that some of these “signals” are not due to drift or noise. The quantitative data derived from the smoothed signals is essential for the discussion in the manuscript and thus might easily lead to inaccurate conclusions. The control experiments to prove the stability of the aptamer modified sensor and the lack of non-specific adsorption are missing, although they were suggested. While recognizing the difficulties in performing more experiments due to Covid situation, additional experiments are necessary as is the and careful reconsideration of the information presented, before recommending the manuscript for publication in Sensors.

Specific comments

-Overlap with work in ref 17: the curve in Figure 3 showing the changes in frequency as a function of the concentration of Listeria innocua solution is a repeat of Figure3 in ref 17 with an extra datapoint added. Nonetheless the Kd derived from the fitting of the data according to the Langmuir isotherm is exactly for both figures. Is this a mistake? As a suggestion to the authors: carefully check any further overlap with ref 17, mention from the beginning the main findings reported in ref 17 as a starting point (e.g. results such as Kd, detection limit, effect of high concentrations of bacteria ) and emphasize even more the focus on the viscoelastic properties of the aptamer layer in this new work. Add new data, in particular control experiments that would be helpful in the interpretation of the changes in the viscoelastic properties of the aptamer layers.

-Sensor stability, drift: from figure 1A it would seem that the drift is indeed “minimal” (as per authors’ response to the previous comments). However one should consider the different scales in figure 1A and 2. I maintain the opinion that the lack of drift should have been evaluated for the 360 min experiment to prove the sensor’s stability in buffer and to validate the very small changes in frequency upon addition of Listeria bacteria shown in figure 2 and further in figure 3.

- Previous comment: Statistical evidence is needed to show that the changes induced following the addition of bacteria solutions of different concentrations are relevant. Authors’ answer: “Each frequency and dissipation values were extracted from at least 3 independent experiments and the standard deviation has been determined”. I reiterate my comment: the authors should provide statistical evidence that the change in frequency after the addition of e.g., 5x103 cfu/mL bacteria is significantly higher (different) than the noise. The noise seems very high in most experiments (around 2 Hz in Figure 2A and B).

- The authors recognized the necessity to perform control experiments, to prove the specificity of the aptasensor but have not included them in the revised manuscript. Instead, they mentioned future plans to explore the potential- non-specific adsorption effects on neutravidin facilitated by extracellular polymeric substance (EPS) from bacteria. However, the aptamer used in this work is known to recognize (in its free form, in solution) other bacteria with similar affinity as Listeria monocua. Therefore, it seems important to investigate non-specific adsorption effects using lower concentrations than 1x105 cfu/mL solutions of E coli or other bacteria. These new results should be added in the main text to help the discussion of non-specific changes in the viscoelastic properties of the aptamer layers, investigated exclusively with large concentrations of E coli.

-Minor grammar and spelling issues:

Line  100-101 should be corrected” So far the viscoelastic properties …has  been studies using …”.

Lines 125-126” French” should be corrected to “France”

Line 207: correct “apatamer”

Author Response

Comment: The quality of the manuscript has significantly improved after the revision however there are some important points still not addressed and some remaining concerns. The main concerns are: 1) there is still too much overlap with authors’ previous work reported in reference 17. 2). The noise seems very high in most experiments. Since the raw signals are very small, and there is no convincing proof that some of these “signals” are not due to drift or noise. The quantitative data derived from the smoothed signals is essential for the discussion in the manuscript and thus might easily lead to inaccurate conclusions. The control experiments to prove the stability of the aptamer modified sensor and the lack of non-specific adsorption are missing, although they were suggested. While recognizing the difficulties in performing more experiments due to Covid situation, additional experiments are necessary as is the and careful reconsideration of the information presented, before recommending the manuscript for publication in Sensors.

Response: We are grateful to the reviewer for additional important comments. We performed additional experiments as recommended and reduced overlaps with article [17].

Specific comments                                                                     

Comment: Overlap with work in ref 17: the curve in Figure 3 showing the changes in frequency as a function of the concentration of Listeria innocua solution is a repeat of Figure3 in ref 17 with an extra datapoint added. Nonetheless the Kd derived from the fitting of the data according to the Langmuir isotherm is exactly for both figures. Is this a mistake? As a suggestion to the authors: carefully check any further overlap with ref 17, mention from the beginning the main findings reported in ref 17 as a starting point (e.g. results such as Kd, detection limit, effect of high concentrations of bacteria) and emphasize even more the focus on the viscoelastic properties of the aptamer layer in this new work. Add new data, in particular control experiments that would be helpful in the interpretation of the changes in the viscoelastic properties of the aptamer layers.

Response: We reduced overlaps with Ref. [17]. In particularly, Figure 3 has been removed and additional experiments were performed confirming the stability of the aptamer sensor in long lasting experiments. New Figure 1 has been included instead of those overlapping with ref 17. We also confirmed non-significant interaction of Listeria innocua and E. coli with neutravidin layer (Figures 2 in main text for Listeria innocua and Figure S1 in Supplementary data for E. coli) as well as non-significant interaction of E. coli with aptasensor. In later case, we extended the concentration range of addition E. coli since 102 to 106 CFU/mL (Figure 4 in the main text). We are also apologizing for mistake in presenting Δfmax and KD values. The correct values are: Δfmax=-7.54 ±0.52 Hz, KD=(5.41±1.51) x104 CFU/mL.

Comment: Sensor stability, drift: from figure 1A it would seem that the drift is indeed “minimal” (as per authors’ response to the previous comments). However, one should consider the different scales in figure 1A and 2. I maintain the opinion that the lack of drift should have been evaluated for the 360 min experiment to prove the sensor’s stability in buffer and to validate the very small changes in frequency upon addition of Listeria bacteria shown in figure 2 and further in figure 3.

Response: We performed additional experiments in which confirmed the small drift of the sensor during long time measurements for 360 min. The new Figure 1 has been added in the main text.

Comment: Previous comment: Statistical evidence is needed to show that the changes induced following the addition of bacteria solutions of different concentrations are relevant. Authors’ answer: “Each frequency and dissipation values were extracted from at least 3 independent experiments and the standard deviation has been determined”. I reiterate my comment: the authors should provide statistical evidence that the change in frequency after the addition of e.g., 5x103 cfu/mL bacteria is significantly higher (different) than the noise. The noise seems very high in most experiments (around 2 Hz in Figure 2A and B).

Response: We performed statistical analysis in which compared the changes in the frequency obtained by averaging of three frequency changes following addition of Listeria innocua at minimal (5x103 CFU/mL) and maximal concentrations (106 CFU/mL). These values were statistically significant according to the Student's t-test with p<0.001. Moreover, as it can be seen from the new Figure 4, addition of E. coli did not result in significant changes of the frequency even at rather high concentration of this bacteria (106 CFU/mL).

Comment: The authors recognized the necessity to perform control experiments, to prove the specificity of the aptasensor but have not included them in the revised manuscript. Instead, they mentioned future plans to explore the potential- non-specific adsorption effects on neutravidin facilitated by extracellular polymeric substance (EPS) from bacteria. However, the aptamer used in this work is known to recognize (in its free form, in solution) other bacteria with similar affinity as Listeria innocua. Therefore, it seems important to investigate non-specific adsorption effects using lower concentrations than 1x105 cfu/mL solutions of E coli or other bacteria. These new results should be added in the main text to help the discussion of non-specific changes in the viscoelastic properties of the aptamer layers, investigated exclusively with large concentrations of E coli.

Response: Additional experiments on the interaction of Listeria innocua and E. coli with neutravidin layers and those of interaction of E. coli with aptamer layers were performed as requested by reviewer and showed negligible effect on the frequency and dissipation. These results have been included in the main text of the revised manuscript.

Comment: Minor grammar and spelling issues:

Line  100-101 should be corrected” So far the viscoelastic properties …has  been studies using …”.

Lines 125-126” French” should be corrected to “France”

Line 207: correct “apatamer”

Response: Thank you for careful reading of the manuscript. All grammar and spelling issue were corrected as recommended.

Reviewer 3 Report

I am happy with the corrections made in the manuscript which can now be published.

The only suggestion I have is to clarify the data acquisition protocol in sensing experiments which requires averaging the raw data prior the evaluation of bacteria concentration. 

Author Response

I am happy with the corrections made in the manuscript which can now be published.

The only suggestion I have is to clarify the data acquisition protocol in sensing experiments which requires averaging the raw data prior the evaluation of bacteria concentration. 

Response: We are grateful to the reviewer for positive opinion on the revised manuscript and for additional useful comment. The acquisition protocol was based on measurement of the changes in frequency and dissipation following addition of bacteria. The experiments were repeated at least 3 times at independently prepared sensors. For construction of the calibration curve the mean and SD were determined for each concentration of bacteria.

            However, according to the suggestions of 1st reviewer and in order to reduce overlaps with previously published data in conference proceedings (reference [17]), we removed calibration plot and analysis of the data by Langmuir isotherm and focused more on the evaluation of the viscoelastic properties of the sensing layers. We also performed additional experiments that evidence on negligible interaction of Listeria innocua and E. coli with neutravidin layers. We also performed additional experiments on interaction of E. coli with aptamers using wider range of concentrations and confirmed that this bacterium did not change significantly the resonant frequency.

Reviewer 4 Report

'Cause most of issues raised by reviewer were resolved, the paper can be acceptable surely.

Author Response

Comment: Cause most of issues raised by reviewer were resolved, the paper can be acceptable surely.

 Response: We are grateful to the reviewer for positive opinion on the revised manuscript.